# Atraric Acid Exhibits Anti-Inflammatory Effect in Lipopolysaccharide-Stimulated RAW264.7 Cells and Mouse Models

**DOI:** 10.3390/ijms21197070

**Published:** 2020-09-25

**Authors:** Seul-Ki Mun, Kyung-Yun Kang, Ho-Yeol Jang, Yun-Ho Hwang, Seong-Gyeol Hong, Su-Jin Kim, Hyun-Wook Cho, Dong-Jo Chang, Jae-Seoun Hur, Sung-Tae Yee

**Affiliations:** 1Department of Pharmacy, Sunchon National University, 255 Jungang-Ro, Suncheon 549-742, Korea; motomoto1210@naver.com (S.-K.M.); hyh7733@naver.com (Y.-H.H.); hong9217@naver.com (S.-G.H.); ksz1353@naver.com (S.-J.K.); djchang@scnu.ac.kr (D.-J.C.); 2Suncheon Research Center for National Medicines, Suncheon 549-742, Korea; kang8404@nate.com (K.-Y.K.); yeol2686@naver.com (H.-Y.J.); 3Department of Biology, Sunchon National University, Suncheon 549-742, Korea; hwcho@sunchon.ac.kr; 4Department of Environmental Education, Korea Lichen Research Institute, Sunchon National University, Suncheon 549-742, Korea; jshur1@sunchon.ac.kr

**Keywords:** anti-inflammation, endotoxin shock, atraric acid, lichen, *Heterodermia hypoleuca*

## Abstract

Lichens, composite organisms resulting from the symbiotic association between the fungi and algae, produce a variety of secondary metabolites that exhibit pharmacological activities. This study aimed to investigate the anti-inflammatory activities of the secondary metabolite atraric acid produced by *Heterodermia hypoleuca*. The results confirmed that atraric acid could regulate induced pro-inflammatory cytokine, nitric oxide, prostaglandin E2, induced nitric oxide synthase and cyclooxygenase-2 enzyme expression in lipopolysaccharide (LPS)-stimulated RAW264.7 cells. Meanwhile, atraric acid downregulated the expression of phosphorylated IκB, extracellular signal-regulated kinases (ERK) and nuclear factor kappa B (NFκB) signaling pathway to exhibit anti-inflammatory effects in LPS-stimulated RAW264.7 cells. Based on these results, the anti-inflammatory effect of atraric acid during LPS-induced endotoxin shock in a mouse model was confirmed. In the atraric acid treated-group, cytokine production was decreased in the peritoneum and serum, and each organ damaged by LPS-stimulation was recovered. These results indicate that atraric acid has an anti-inflammatory effect, which may be the underlying molecular mechanism involved in the inactivation of the ERK/NFκB signaling pathway, demonstrating its potential therapeutic value for treating inflammatory diseases.

## 1. Introduction

Inflammation is an inherent immune mechanism that occurs as a result of pathogen invasion in the body [1,2]. During an inflammatory immune response, macrophages release pro-inflammatory mediators, such as nitric oxide (NO), prostaglandin E2 (PGE2), induced nitric oxide synthase (iNOS), cyclooxygenase-2 (COX-2), as well as pro-inflammatory cytokines and chemokines [3,4]. Nuclear factor-kappa B (NFκB), which regulates the transcription of inflammatory factors induced by lipopolysaccharide (LPS) stimulation, binds to IκB in the cytoplasm and activated by phosphorylation and IκB degradation. The activation of NFκB is regulated by a mitogen-activated protein kinase (MAPK) and an extracellular signal-regulated kinase (ERK) [5,6,7]. Abnormally high inflammatory response results in severe tissue damage and endotoxin shock. Chronic inflammatory responses cause a series of diseases such as neurodegeneration, cardiovascular diseases (CVD), and cancer [3]. Endotoxin shock is a severe inflammatory reaction caused by Gram-negative bacteria infection of the bloodstream. LPS, an endotoxin on the surface of Gram-negative organisms, stimulates the release of pro-inflammatory cytokines and NO, resulting in increasing vasodilation and vascular permeability. This leads to impaired cardiovascular function, pulmonary dysfunction, acute renal impairment, septic shock and, finally, death. Many studies have demonstrated that the inhibition of NFκB and MAPK activation is an important strategy for the treatment of inflammatory diseases [8,9,10,11].

Lichens, symbiotic associations of algae and fungi, are abundantly found in the surroundings and can grow in cold and harsh environments [3,12]. In addition, lichens have been traditionally used as herbal tea and as a food ingredient in China and Japan. Their ability capacity to produce and accumulate secondary metabolites gives rise to their wide chemical diversity. Many secondary metabolites from Lichen, including atraric acid, atranorin, usnic acid and olivetoric acid, have been linked to different biological effects such as antibacterial, antioxidant and antiproliferation effects. Therefore, lichens are attracting attention as a new natural material. [13,14,15].

Although biological activities of numerous lichens have been studied, there is no information on *hyterodermia hypoleuca* (HH), a lichen native to Korea. Thus, we attempted to discover the biological activities of HH from the extract fractions using diverse solvents. The methanol (MeOH) extract exhibited a strong anti-inflammatory activity. However, there is no information for anti-inflammatory metabolites of HH, and also the mechanism underlying the anti-inflammatory activity of HH is not yet clear. Thus, we tried to discover active metabolites with anti-inflammatory activity from the MeOH extract fraction. We successfully isolated atraric acid with strong anti-inflammatory activity as a secondary metabolite with high concentration in the MeOH extract. Atraric acid is a phenolic compound, and has been reported to exhibit anti-cancer activity against prostate cancer. It serves as an antagonist of androgen receptor [16]. Atraric acid also possesses antioxidant and antibacterial [17] activities. However, it is not clear how atraric acid shows anti-inflammatory activity in vitro and in vivo. In this work, we show the in vitro and in vivo anti-inflammatory effects of atraric acid, which revealed that atratic acid has anti-inflammatory activity caused by the suppression of the MAPK-NFκB signaling pathway.

## 2. Results

### 2.1. HH Exhibited Anti-Inflammatory Effect

The anti-inflammatory activity of *heterodermia hypoleuca* (HH) was investigated in LPS-stimulated RAW264.7 cells. At first, examining the cytotoxicity of HH extract in RAW264.7 cells using cell counting kit-8 (CCK-8), we observed that the growth of the cells was not affected by the HH extracts (Figure 1a). As shown in Figure 1, HH inhibited the production of NO stimulated by LPS in a concentration-dependent manner (Figure 1b). In addition, the levels of pro-inflammatory, such as Tumor necrosis factor-alpha (TNF-α), interleukin l beta (IL-1β), interleukin 6 (IL-6) and granulocyte–macrophage colony-stimulating factor (GM-CSF) released in LPS-stimulated RAW 264.7 cells were significantly reduced by 20%, 30%, 58% and 45%, respectively, at 30 μg/mL of HH (Figure 1c–f).

### 2.2. Identification of Atraric Acid

The yield and purification of atraric acid attained by the high performance liquid chromatography (HPLC) purification (Appendix A) of MeOH extract of HH (Appendix A) are 2.05% and 96.5% respectively. The structure of atraric acid was identified using nuclear magnetic resonance (NMR) and mass spectroscopy (MS) and was compared to the previously reported data. Atraric acid: 1H NMR (methanol-d4, 400 MHz); 6.25 (1H, s, H-6), 3.83 (3H, s, H-10), 2.31 (3H, s, H-8), 1.89 (3H, s, H-9), ^13^C NMR (methanol-d4, 100 MHz); 172.5 (C-1), 162.1 (C-3), 160.1 (C-5), 140.0 (C-7), 111.1 (C-6), 108.4 (C-4), 104.4 (C-2), 52.4 (C-10), 23.9 (C-8), 8.1 (C-9); Electrospray ionization (ESI)-MS showed an [M+H]^+^ ion at m/z = 197.0 [M+H]^+^, 219.0 [M+Na]^+^ (The number of carbone in ^13^C NMR chemical shifts was identified using 2D NMR data and confirmed by comparing them with previously reported data) (Figure 2) [18,19,20].

### 2.3. Atraric Acid Inhibited the Production of NO and Pro-Inflammatory Cytokines

We investigated the production of NO, PGE2 and pro-inflammatory cytokines, such as IL-1β, IL-6 and GM-CSF, in LPS-stimulated RAW264.7 cells to determine the anti-inflammatory activity of atraric acid isolated from HH. Based on the cell viability test, we performed the in vitro test of atraric acid with concentrations of 1 to 300 μM for the evaluation of the anti-inflammatory activity. As shown in Figure 3, we confirmed that atraric acid did not affect the viability of RAW264.7 cells at a concentration ranging from 1 to 300 μM. The levels of NO, PGE2 and pro-inflammatory cytokines were determined after the treatment of the LPS-stimulated RAW264.7 cells with atraric acid. LPS stimulation significantly increased the production of NO, PGE2 and all pro-inflammatory cytokines in RAW264.7 cells. The Griess reaction showed that atraric acid significantly reduced the LPS-stimulated NO production in a dose-dependent manner (69% of inhibition at 300 μM, Figure 4a). The pro-inflammatory cytokines in the culture media were measured using an enzyme-linked immunosorbent assay (ELISA) kit. Treatment of the LPS-stimulated RAW264.7 cells with atraric acid resulted in a dose-dependent decrease in PGE2, IL-1β, IL-6 and GM-CSF levels induced by LPS (100%, 36.3%, 93.3% and 80.9% of inhibition at 300 μM respectively). The results clearly revealed that atraric acid exhibits anti-inflammatory activity by regulating the levels of inflammatory factors produced in LPS-stimulated RAW264.7 cells.

### 2.4. Atraric Acid Inhibited LPS-Induced Expression of iNOS and COX-2

iNOS and COX-2 are pro-inflammatory proteins that are key enzymes in catalyzing the production of NO and PEG2, respectively [21]. To investigate the effect of atraric acid on the production of NO and PGE2, the expressions of the enzymes iNOS and COX-2 were evaluated using Western blot and flow cytometry. Our result showed that in LPS-stimulated RAW264.7 cells, expression of iNOS and COX-2 was significantly enhanced, whereas treatment with atraric acid resulted in decreased expression of iNOS and COX-2 in a concentration-dependent manner (Figure 5).

### 2.5. Atraric Acid Suppressed LPS-Stimulated Phosphorylation of the Nfκb Signaling Pathway

The NFκB signaling pathway holds significant importance in the inflammation process. NFκB exists in the inactive state bind to IκB which is an inhibitor of NFκB. However, an inflammatory stimulation triggers the LPS-induced phosphorylation of IκB followed by its degradation. The NFκB is then phosphorylated and converted to its active form. In the NFκB signaling pathway, MAPK regulates the activation of NFκB; the phosphorylation of MAPK induces the activation of NFκB followed by subsequent expression of inflammatory mediators and pro-inflammatory cytokines [10,22]. To investigate the effect of atraric acid on the MAPK/NFκB signaling pathway, LPS-stimulated RAW264.7 cells were treated with atraric acid to analyze the expression of the inflammatory mediators by Western blot. Figure 6 showed that atraric acid reduced the expression of p-IκB, which resulted in the degradation of IκB, and p-ERK in LPS-stimulated RAW264.7 cells. The result showed that the phosphorylation of NFκB was inhibited by the treatment with atraric acid (Figure 6e), implying the potential value of atraric acid as an anti-inflammatory agent.

### 2.6. Atraric Acid Exhibited Anti-Inflammatory Effects on LPS-Induced Endotoxin Shock in Mice

Based on in vitro study for the anti-inflammatory activity of atraric acid, in vivo efficacy of atraric acid was evaluated on LPS-induced endotoxin shock in mice. Endotoxin shock not only increases the level of cytokines in the blood, but also induces tissue damage, causing secondary disease [23,24]. The inhibitory effect of atraric acid on the release of inflammatory cytokines from the serum and peritoneal lavage fluid was first investigated. As shown in Figure 7, the LPS-stimulated group of mice exhibited significantly increased production of pro-inflammatory cytokines, whereas, for ataric acid-treated mice, the production of pro-inflammatory cytokines was inhibited (Figure 7b-e). Next, we performed hematoxylin and eosin (H&E) staining to analyze the pathological characteristics of the organs, including the kidney, liver, and lung. In the LPS group, vasodilation in glomerular atrophy, bleeding, and recruitment of inflammatory cells were observed in the kidney, and similar pathological damage such as vasodilation and bleeding were observed in the liver. In addition, the alveolar septa became thicker and the alveoli exuded, leading to the destruction of some alveolar structures in the lung of the LPS-stimulated mice (Figure 7f). On the contrary, the atraric acid-treated group of mice exhibited reduced pathological damages such as vasodilation and bleeding. Taken together, it was confirmed that atraric acid suppresses the inflammatory response induced by LPS in vitro and in vivo.

## 3. Discussion

There is a great interest in the discovery of biologically active natural products due to their activities with relatively low toxicity as compared to other modalities such as synthetic chemicals [10]. Numerous studies have shown that natural products are developed as beneficial dietary supplements for health and therapeutic agents against disease [25]. Lichen is a unique organism that produces biologically active secondary metabolites with various pharmacological effects including antioxidant, anti-cancer [17] and anti-viral effects [16]. In the LPS stimulated inflammation, toll-like receptor-4 (TLR4) and MD-2 in macrophages form heterodimers that recognize common patterns of an LPS molecule [26]. Upstream signaling initiated from the interaction of LPS and TLR4 complex induced the activation of MAPKs/NFκB signaling pathway, which results in overexpression of inflammatory factors [22]. The overexpression of the inflammatory factors, such as enzymes iNOS and COX-2 and pro-inflammatory cytokines including TNF-α, IL-6, and IL-1β, induces cellular inflammatory responses and consequently leads to in vivo damage such as endotoxin shock (Figure 8) [6,21]. LPS-stimulated macrophages have been generally used as an in vitro model to evaluate the anti-inflammatory activity of numerous natural products and synthetic compounds.

The purpose of this study was to discover an anti-inflammatory agent from a natural product. The same was successfully achieved by isolation and in vitro/in vivo evaluation of secondary metabolites obtained from a natural source; that is, lichens. Atraric acid has been previously reported to have various biological activities such as antibacterial and anti-cancer activity [16,17]. However, we first isolated the atraric acid from *hyterodermia hypoleuca*, and reported its anti-inflammatory activity in vitro and in vivo. In this study, we first screened several fractions of solvent extracts of *heterodarmia hypoleuca* by estimating inhibitory activities for the production of inflammatory factors including NO, TNF-α, IL-6, IL-1β, and GM-CSF in LPS-stimulated macrophages, which revealed that the methanol extracts showed the most potent anti-inflammatory activity (Figure 1). Next, we separated and purified atraric acid from the MeoH extract using HPLC (Figure 2, Appendix A). The isolated atraric acid exhibited a strong inhibition for the production of NO, PGE2, IL-6, IL-1β and GM-CSF in LPS-stimulated RAW264.7 cells (Figure 4). Atraric acid also reduced the protein expression of enzymes iNOS and COX-2, involved in the production of inflammatory mediators such as NO and PGE2. In order to prove the anti-inflammatory effect of atraric acid in vivo, the effect of atraric acid was evaluated with LPS-induced endotoxin shock in mice. Endotoxin shock was induced by the treatment with LPS (10 mg/kg i.p. injection) in mice. Two different concentrations of atraric acid (10 and 30 mg/kg) were injected 2 h before and 4 h after LPS administration. Excessive pro-inflammatory cytokines in the serum and peritoneum lavage were observed due to the systematic inflammatory reactions in the LPS-stimulated groups. Furthermore, the LPS-stimulated mice also exhibited histological signs of necrosis resulting from vasodilation and bleeding of the organs. However, in the atraric acid-treatment mice, the LPS-induced pro-inflammatory cytokine expression was inhibited. Histological staining in these mice showed reduced vasodilation and bleeding of the damaged organs (Figure 7b–f). The in vitro and in vivo data demonstrate that atraric acid can be a promising therapeutic agent against inflammatory diseases.

## 4. Materials and Methods

### 4.1. Collection and Preparation of the Lichen

*Heterodermia hypoleuca* (*Kol.170037*, HH) was provided by Allied Bioresource Center in the Korean Lichen Research Institute, Sunchon National University, Korea. HH was collected from the coastal rocks of the southern part of Korea. The dried Lichen thalli (60 g) were extracted with 2 L methanol (MeOH) at room temperature for 48 h using sonication. The extract was then filtered and concentrated under vacuum at 40 °C using a rotary evaporator.

### 4.2. Chemicals and Reagents

Roswell park memorial institute-1640 medium (RPMI 1640) and fetal bovine serum (FBS) were purchased from Hyclone Laboratories (Hyclone, South Logan, UT, USA). Cell counting kit–8 (CCK-8) was purchased from Dojindo Laboratories (Dojindo, Kumamoto, Japan). Dimethyl sulfoxide (DMSO), bovine serum albumin (BSA) and Lipopolysaccharides (LPS) were purchased from Sigma Aldrich (St. Louis, MO, USA). Purified rat anti-mouse (TNF-α, IL-6, IL-10, IL-1β, GM-CSF) and biotin rat anti-mouse (TNF-α, IL-6, IL-10, IL-1β and GM-CSF) were purchased from BD Biosciences (San Diego, CA, USA). PGE2 ELISA kit was purchased from R&D Systems (R&D, Minneapolis, MN, USA). HPLC grade acetonitrile, water and methanol were purchased from J.T Baker (USA), while ODS-A (40–60 mesh), Luna 5u (C_18_ 100A 250 × 110 nm), and YMC-actus (Triat C_18_ 100A 250 × 20 mm) were from Merck Co. (Germany), Phenomenex Inc (USA), and YMC (Japan), respectively. Atraric acid, as standard component, was obtained from ChemFaces Biochemical Co., Ltd. (Wuhan, China). The purity of atraric acid (over 96.5%) that was isolated from HH in our laboratory was determined by a high-performance liquid chromatography-evaporative light scattering detector (Appendix A).

### 4.3. Isolation and Analysis

HH extract was dissolved in ethyl acetate (EtOAc) and was partitioned with water (30% MeOH). EtOAc fraction dissolved in MeOH (50% DMSO) was separated by preparative liquid chromatography (Prep-LC, YMC-actus Triart C_18_ 250 × 20 nm column, YMC, Japan), 15 mL/min, UV detection at 254, 365 nm using gradient elution 5% solvent B to 100% solvent B as eluent to afford 63 fractions (1‒63) (solvent A: Water + formic acid 0.5%, solvent B: Acetonitrile + formic acid 0.5%). Atraric acid (39.5 g, Rt = 36.9 min) was obtained by reversed-phase HPLC (Luna 5u C_18_ 100A 250 × 20 nm column, Phenomenex Inc.,USA), 1.0 mL/min, UV detections at 254 nm using isocratic elution, solvent B (Acetonitrile + formic acid 0.5%): 0–60 min; as eluent 50%.

LC-MS/ELSD analysis was performed to analyze the purified atraric acid. The column used for the analysis was a Phenomenex C_18_ (5 µm, 4.6 × 250 mm) column. As analysis conditions, the initial solvent 5% solvent B hold at for 5 min. The gradient condition, Solvent B was then increased from 5% to 70% within 40 min, following proportions solvent B; 45–50 min, 70–100%, hold at B; 100% for 10 min. Solvent A and B in 0.1% Trifluoroacetic acid (TFA).

### 4.4. Instruments and Data Collection

NMR spectra were recorded on a JNM-ECZS series FT 400 NMR (JEOL, Japan) spectrometer and Varian Inova spectrometers using DMSO-d_6_ as the solvent, obtained from Cambridge Isotope Laboratories (CIL), Inc. Chemical shifts were confirmed to be Atraric acid compared with the reported studies. Mass spectra were analyzed on Agilent Technologies 6120 Quadrupole mass spectrometer coupled with an Agilent Technologies 1260 series HPLC, Evaporative light scattering detector (ELSD) G4260A and Electrospray ionization source (ESI) low resolution.

### 4.5. Cell Culture

RAW264.7 cells (murine macrophage cell line, KCLB 40071) were purchased from Korean Cell Line Bank (Seoul, South Korea). The cells were grown in RPMI 1640 supplemented with 10% FBS, 100 units/mL of penicillin, 100 μg/mL of streptomycin (Invitrogen, Carlsbad, CA, USA), and 2-mercaptoethanol (50 μM) in a humidified atmosphere at 37 °C with 5% CO_2_

### 4.6. Cell Viability Assay

Cell viability was determined by CCK-8 assay. RAW 264.7 cells (5 × 10^4^ cell/well) were seeded in a 96-well plate and incubated over-night before experimental interventions. Next, the cells were treated with different concentrations of samples for 24 h. Thereafter, 10 μL of CCK-8 was added to each plated and incubated for 2 h at 37 °C with 5% CO_2_. The optical density was then read at 450 nm using a microplate reader (Versa Max, molecular devices, Sunnyvale, CA). The cell viability was evaluated by comparing the absorbance values of the sample groups with that of the control group, which was considered as 100%.

### 4.7. Measurement of NO and Cytokines

The concentrations of NO, TNF-α, IL-6 IL-1β and GM-CSF were measured by Griess assay and ELISA assay. RAW 264.7 cells (5 × 10^5^ cell/mL) were seeded in a 24-well plate and incubated over-night before experimental interventions. The cells were treated with various concentrations of sample extracts in the presence of LPS (1 μg/mL) for 24 h at 37 °C with 5% CO_2_. Subsequently, the culture supernatant was assayed according to the manufacturer’s instructions [9,27].

### 4.8. Western Blot Analysis

The cell was collected and washed twice with cold phosphate-buffer saline (PBS). Total protein was extracted by using radioimmunoprecipitation assay buffer (Thermo, Rockford, IL, USA) in the presence of protease and phosphatase inhibitor cocktail (Thermo, Rockford, IL, USA). Protein concentration was determined using the BCA protein assay kit (Thermo, Rockford, IL, USA). Equal amounts of protein were separated by 4–12% bis-tris plus gels (Thermo, Rockford, IL, USA) and transferred to nitrocellulose membranes (Thermo, Rockford, IL, USA). The membranes were incubated with blocking solution for 1 h at room temperature followed by overnight incubation at 4 °C with primary antibodies. The primary antibodies included specific antibody β-actin (1:1000, Thermo, Rockford, IL, USA), iNOS (1:500, Thermo, Rockford, IL, USA), COX-2 (1:1000, Thermo, Rockford, IL, USA), IκB (1:1000, Thermo, Rockford, IL, USA), p-IκB (1:1000, Thermo, Rockford, IL, USA), NFκB (1:1000, Thermo, Rockford, IL, USA) and p-NFκB (1:1000, Thermo, Rockford, IL, USA). The membranes were washed with TBST and incubated in horseradish peroxidase (HRP)-conjugated secondary antibody (1:1000, Thermo, Rockford, IL, USA) for 1 h at room temperature while shaking. Next, the membranes were washed with TBST and developed with the enhanced chemiluminescence kit (Thermo, Rockford, IL, USA). The protein bands were captured and measured using a bio-imaging system (Microchemi 4.2 Chemilumineszenz-system, Neve Yamin, Israel) [28].

### 4.9. Animals and Experimental Design

Female BALB/c mice (7 weeks old), weighing about 17‒20 g, were purchased from orient-bio (Orientbio Inc., Seongnam, Korea).

The animals were housed in a controlled environment [22 ± 2 and 50 ± 5% (relative humidity)] in polycarbonate cages and fed a standard animal diet with water. All mice were treated strictly according to the guidelines for laboratory animal care and use issued by the Sunchon National University Institutional Animal Care and Use Committee (SCNU IACUC). All procedures were approved by SCNU IACUC (license number: SCNU IACUC-2019-15, approval date: 12 Dec 2019), and all the experiments performed on mice were performed with extra care and concern.

For the endotoxin shock model, a total of 20 mice were randomly divided into a control group, LPS (10 mg/kg, i.p.) group, and atraric acid-treated group (10 mg/kg, 30 mg/kg, i.p.) (*n* = 5). As shown in Figure 7a, atraric acid was treatment was conducted 2 h prior to LPS administration standard, and atraric acid was intraperitoneally administered to mice for 4 h after LPS injection. Eight hours after the last administration, blood samples were taken, serum samples were prepared for pro-inflammatory cytokine measurement, and organs (kidney, liver, lung, spleen) were extracted for tissue staining [26,29].

### 4.10. Histopathological Examination

Tissues were fixed in 4% formalin and embedded in paraffin. Sections of 4 μm thickness were obtained and stained with H&E and histological changes were monitored under the microscope at × 400 magnification [10,30].

### 4.11. Statistical Analysis

Data are presented as mean ± standard deviation (SD) or standard error of the mean (SEM). The statistical differences between groups were analyzed by one-way SPSS version 22 (SPSS, Chicago, IL, USA) followed by Student’s t-test. A *p*-value of 0.05 or less indicated statistical significance.

## 5. Conclusions

Atraric acid isolated from the MeOH extract of *hyterodermia hypoleuca* significantly reduced the production of NO, PGE2, TNF-α and IL-6 in LPS-induced RAW 264.7 macrophages and suppressed the expression of enzymes iNOS, COX-2. In addition, the inhibition of NFκB/ERK in NFκB-MAPK pathways, which is one of the inflammatory signaling pathways, was confirmed in vitro. Based on in vitro results, the in vivo anti-inflammatory effect of atraric acid was evaluated in the LPS-induced mouse endotoxin shock model, showing the downregulation of inflammatory cytokines by LPS-induced endotoxin shock and the decrease in vasodilation and bleeding in damaged organs. Taken together, it was confirmed that atraric acid is a potential target compound for the development of a novel anti-inflammatory agent.

## 6. Patents

A patent for an anti-inflammatory composition containing an extract component derived from lichen was filed. [Patent application number: 10-2018-0128880].

## Figures and Tables

**Figure 1 ijms-21-07070-f001:**
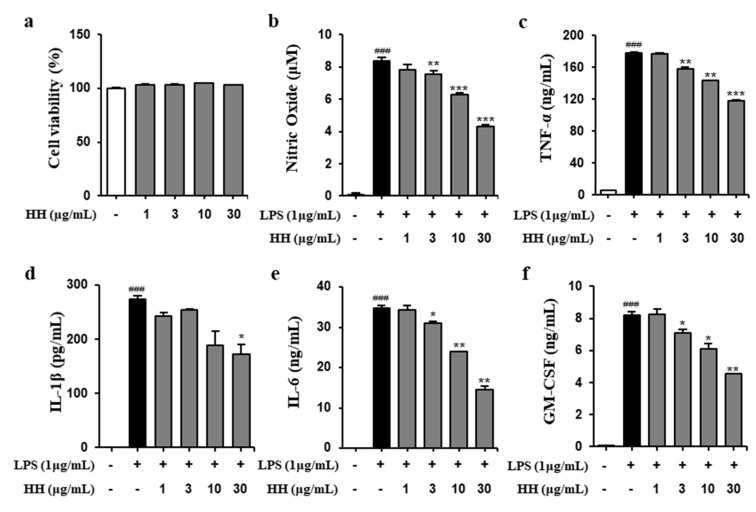
The screening of anti-inflammatory activity of the methanol extract of *Heterodermia hypoleuca* (HH). (**a**) The cells were treated with different concentrations of HH for 24 h, and the viability of the treated cells was determined by cell counting kit-8 assay. The cells were preincubated with 1 μg/mL lipopolysaccharide (LPS) for 1 h and then treated with 1-30 μg/mL HH for 24 h. (**b**) The concentration of nitric oxide in the cultured medium was measured by using the Griess reaction. The release of Tumor necrosis factor-alpha (TNF-α) (**c**), interleukin l beta (IL-1β) (**d**), interleukin 6 (IL-6) (**e**) and granulocyte–macrophage colony-stimulating factor (GM-CSF) (**f**) was determined by enzyme-linked immunosorbent assay (ELISA). Data are presented as mean ± SD from three independent experiments (^###^
*p* < 0.001 versus the control group; * *p* < 0.05, ** *p* < 0.01, *** *p* < 0.001 versus the LPS group).

**Figure 2 ijms-21-07070-f002:**
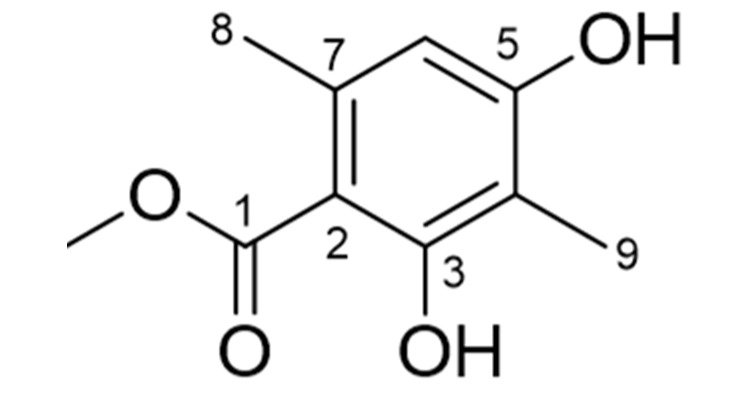
Chemical structure of atraric acid.

**Figure 3 ijms-21-07070-f003:**
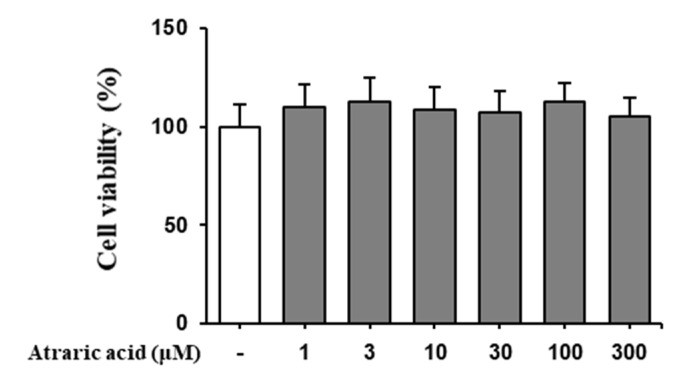
Effect of atraric acid on the viability of RAW264.7 cells. The cells were treated with 1–300 μM atraric acid for 24 h. Cell viability was determined using cell counting kit-8. Data are presented as mean ± SD from three independent experiments.

**Figure 4 ijms-21-07070-f004:**
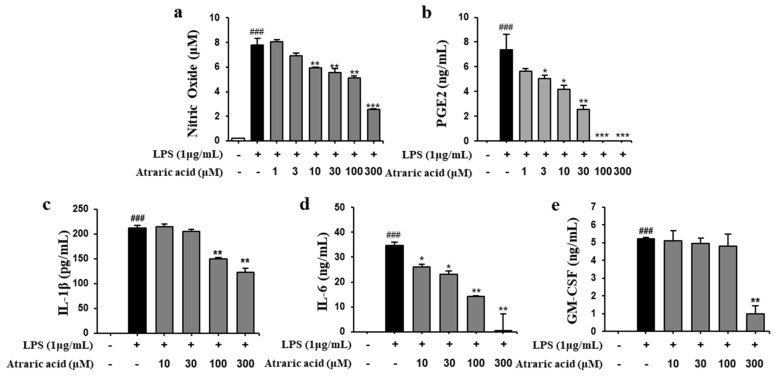
Effect of atraric acid on the levels of nitric oxide (NO), prostaglandin E2 (PGE2) and pro-inflammatory cytokines produced by LPS-stimulated RAW2647 cells. The cells were pretreated with 1 μg/mL LPS for 1 h and subsequently treated with 1–300 μM atraric acid for 24 h. (**a**) The concentration of nitric oxide in the cultured medium was measured as an indicator of NO production by using Griess reaction. The release of PGE2 (**b**), IL-1β (**c**), IL-6 (**d**) and GM-CSF (**e**) was determined by ELISA. Data are presented as mean ± SD from three independent experiments (^###^
*p* < 0.001 versus the control group;* *p* < 0.05, ** *p* < 0.01, *** *p* < 0.001 versus the LPS group).

**Figure 5 ijms-21-07070-f005:**
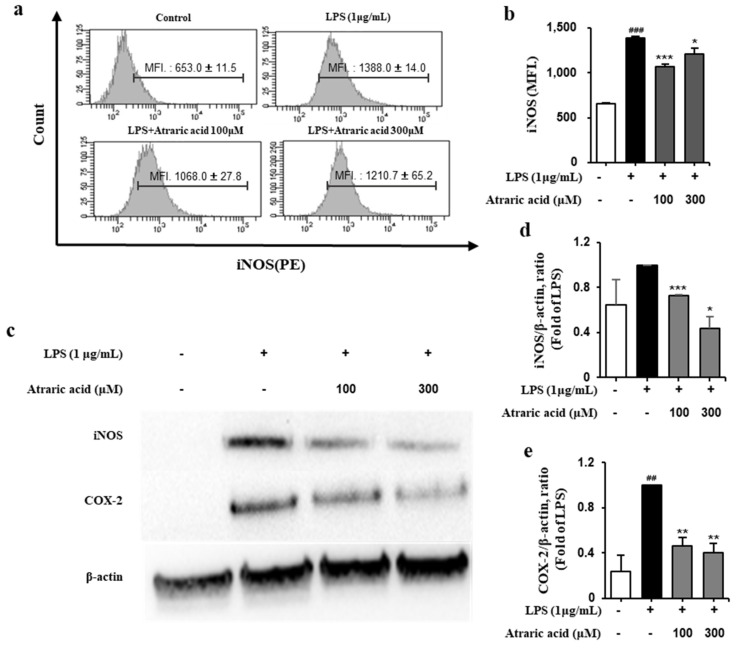
Effect of atraric acid on induced nitric oxide synthase (iNOS) and cyclooxygenase-2 (COX-2) expression in LPS-stimulated RAW2647 cells. The cells were pre-treated with 1 μg/mL LPS for 1 h and subsequently treated with 100, 300 μM atraric acid for 18 h. (**a**) The expression of iNOS was analyzed by flow cytometry using specific antibodies and isotype controls. (**b**) Mean fluorescence intensities (MFI.) are shown for each panel. (**c**) Western blot analysis was performed to investigate the effect of atraric acid on the expression of pro-inflammatory proteins. The ratios of iNOS (**d**) and COX-2 (**e**) to β-actin after atraric acid treatment were determined, respectively. Data are presented as mean ± SD from two independent experiments (^##^
*p* < 0.01, ^###^
*p* < 0.001 versus the control group; * *p* < 0.05, ** *p* < 0.01, *** *p* < 0.001 versus the LPS group).

**Figure 6 ijms-21-07070-f006:**
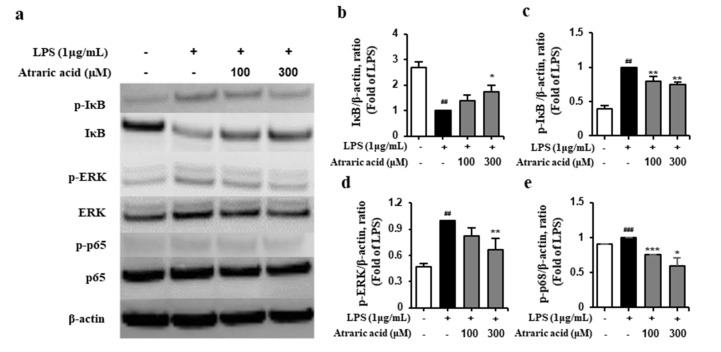
Effects of atraric acid on the activation of NFκB signaling. RAW264.7 cells were pre-incubated with 1 μg/mL LPS for 30 min and then treated with 100, 300 μM atraric acid for 4 h. (**a**) The inhibitory effect of atraric acid against the phosphorylation of IκB, extracellular signal-regulated kinase (ERK), and NκB was detected by Western blot. The ratios of degradation of IκB (**b**), p-IκB (**c**), p-ERK (**d**) and p-NFκB (**e**) to β-actin after atraric acid treatment were determined, respectively. Data are presented as mean ± SD from two independent experiments (^##^
*p* < 0.01, ^###^
*p* < 0.001 versus the control group; * *p* < 0.05, ** *p* < 0.01, *** *p* < 0.001 versus the LPS group).

**Figure 7 ijms-21-07070-f007:**
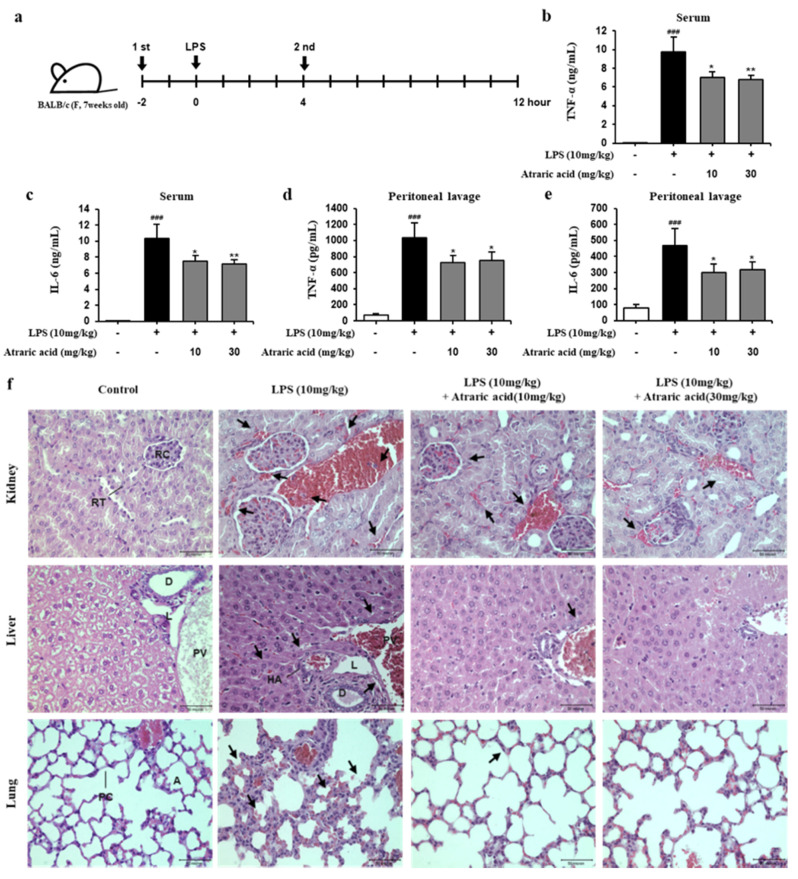
Effects of atraric acid on LPS-induced endotoxin shock in mice. (**a**) BALB/c mice (*n* = 5) were intraperitoneally administered (i.p.) LPS (10 mg/kg, *E*. *coli* 011:B4) and subsequently treated with atraric acid (10 mg/kg, 30 mg/kg) or control (PBS) for 4 h. (**b**–**e**) The levels of pro-inflammatory cytokines released by the serum and peritoneal lavage were detected by ELISA assay. (**f**) Tissue samples were stained with H&E to exhibited histopathological changes (Abbreviations: RT, renal tubule; RC, renal corpuscle; D, bile duct; L, lymphatic vessels; PV, portal vein; HA, hepatic artery; PC, pulmonary capillaries; A, alveoli). The arrows showed inflammatory infiltration. Data are presented as mean ± SEM from five independent experiments (^###^
*p* < 0.001 versus the control group; * *p* < 0.05, ** *p* < 0.01 versus the LPS group) Scale bar: 50 μm.

**Figure 8 ijms-21-07070-f008:**
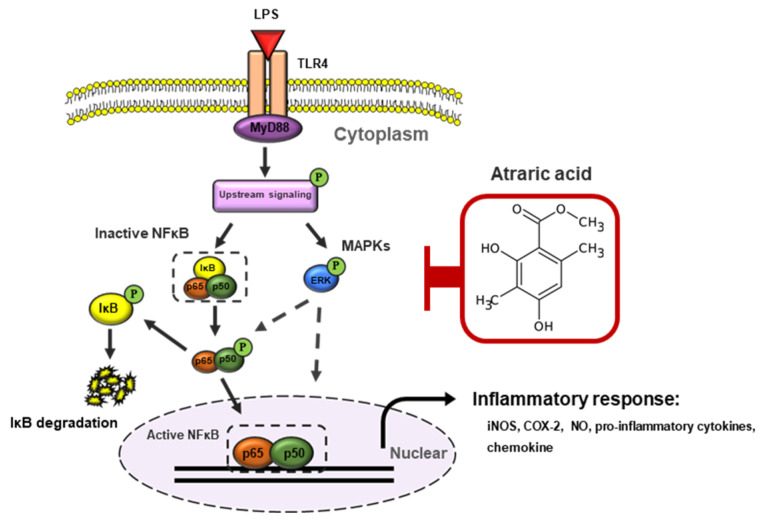
Schematic representation of the anti-inflammatory effects of atraric acid in RAW264.7 cells. Abbreviations: LPS, lipopolysaccharide; TLR4, Toll-like receptor 4; MyD88, myeloid differentiation factor 88; MAPKs, mitogen-activated protein kinase; ERK, extracellular regulated protein kinase; NFκB, nuclear factor-κB; iNOS, inducible nitric oxide synthase; COX-2, cyclooxygenase-2; NO, nitric oxide.

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
