# Peer review of "Atraric Acid Exhibits Anti-Inflammatory Effect in Lipopolysaccharide-Stimulated RAW264.7 Cells and Mouse Models"

_ijms, 2020, doi:10.3390/ijms21197070_

Round 1
Reviewer 1 Report
The paper by Mun et al. was devoted to investigate both “in vitro” and “in vivo” the anti-inflammatory properties of a methanol extract of Heterodermia hypoleuca and atraric acid. The manuscript is interesting but need some major revision before publication as follow:
The introduction must be improved. The different sections should be linked to each other. For instance: Lines 38-41: it should be explained what NFkB is and why it was taken into account;
Lines 60-61: it should be explained what Heterodermia hypoleuca is and why it was used as a source of atraric acid.
The results section must be improved. Lines 65-66: this sentence is not clear, what is the meaning of “the cell viability of HH extracts”? It should be explained how the HH extract was obtained and what it contains. How much atraric acid is contained in the HH extract?
Lines 81-87, the purity of atraric acid (%) should be declared and analytically demonstrated and showed.
Lines 119-120: this sentence is not clear, the expression of iNOS and COX-2 proteins were increased by LPS and were decreased by atraric acid.
Lines 136-138: As shown in Figure 6c the expression of p-IκB was not statistically significant reduced by tartaric acid
Lines 159-161: this sentence is not clear “atraric acid treatment…… improved the pathological properties of the organs”
The discussion is poor and must be improved.
Lines 175-178: this sentence is not clear, “its value….” whose? who is the subject of this sentence?
Lines 179-182: this sentence is not clear, perhaps it should be divided into two sentences
Line 188: the data must be show (see above).
Figure 8 is never mentioned in the text.
Author Response
Q1. Lines 38-41: it should be explained what NFκB is and why it was taken into account.
A1: According to your opinion, we added the relationship between NFkB and inflammation, and thereby corrected two sentences of introduction on line 38-41 (”Nuclear factor kappa B (NFκB) is inactivated………..and an extracellular regulatory kinase [5-7 ]”) as following,
“Nuclear factor kappa B (NFκB), which regulates the transcription of inflammatory factors induced by LPS stimulation, binds to IκB in the cytoplasm and is activated by phosphorylation followed by IκB degradation. The activation of NFκB is regulated by mitogen-activated protein kinase (MAPK) and extracellular regulatory kinase [5-7].”
Q2. Lines 60-61: it should be explained what Heterodermia hypoleuca is and why it was used as a source of atraric acid. The results section must be improved.
A2: Hyterodermia hypoleuca is a lichen native to Korea and there is no any biological study for hyterodermia hypoleuca. Thus, we have tried to find a biological activity of hyterodermia hypoleuca, and found that it has a anti-inflammatory effect. However, there is no information for anti-inflammatory metabolites of hyterodermia hypoleuca, and also its mode of action for anti-inflammatory activity is not clear. Thus, we had tried to discover active metabolites with anti-inflammatory activity from the extract fractions using diverse solvents followed by HPLC purification. As a result, atraric acid was discovered from MeOH extract and it showed a strong anti-inflammatory activity. Therefore, according to your opinion, we have corrected a part (line 58-71) of introduction as following,
“Biological activities of numerous lichens have been studied, but there is no any information for a biological activity of hyterodermia hypoleuca, a lichen native to Korea. Thus, we had tried to discover a biological activity of hyterodermia hypoleuca from the extract fractions using diverse solvents, which revealed the MeOH extract showed a strong anti-inflammatory activity. However, there is no information for anti-inflammatory metabolites of hyterodermia hypoleuca, and also the mechanism for anti-inflammatory activity of hyterodermia hypoleuca was not clear. Thus, we have tried to discover active metabolites with anti-inflammatory activity from the MeOH extract fraction showing high anti-inflammatory activity. As a result, we successfully isolated atraric acid with strong anti-inflammatory activity as a secondary metabolite with high content in the MeOH extract. Atraric acid is a phenolic compound, and has been reported to have anti-cancer activity against prostate cancer as an antagonist of androgen receptor [17] as well as antioxidant and antibacterial [16] activities. But, it is not clear how atraric acid shows anti-inflammatory activity in vitro and in vivo. In this work, we show in vitro and in vivo anti-inflammatory effects of atraric acid, which revealed that atratic acid has the anti-inflammatory activity cause by the suppression of MAPK-NFκB signaling pathway”
Q3. Lines 65-66: this sentence is not clear, what is the meaning of “the cell viability of HH extracts”?
A3: We have corrected the sentence (“the cell viability of HH extracts….”) as following,
“Prior to performing the experiments for anti-inflammatory activity of HH, the cytotoxicity of HH extract was evaluated in RAW264.7 cells using cell counting kit-8 (CCK-8), which showed the growth of the cells was not affected by HH extracts (Figure 1a).”
Additionally, we deleted the sentence in the line 64-66 (“This section may be divided by subheadings. It should provide a concise and precise description of the experimental results, their interpretation as well as the experimental conclusions that can be drawn”)
Q4. It should be explained how the HH extract was obtained and what it contains. How much atraric acid is contained in the HH extract?
A4: We corrected the part of “2.2. Identification of atraric acid”as following,
“Atraric acid with the 2.05% of yield and 96.5% of purity was isolated from the methanol extract of HH (Figure S1) by HPLC purification (Figure S2, Table S1).”
Q5. Lines 81-87, the purity of atraric acid (%) should be declared and analytically demonstrated and showed.
A5: HPLC analysis showed that the purity of the atraric acid from the MeOH extract is 96.5%, and the HPLC chromatograms for separation of atraric acid was attached in supplementary data (Figure S2, Table S1).
Q6. Lines 119-120: this sentence is not clear, the expression of iNOS and COX-2 proteins were increased by LPS and were decreased by atraric acid.
A6: In order to remove the confusion from the pointed sentence, we corrected the whole paragraph in “2.4. Atraric acid inhibited LPS-induced expression of iNOS and COX-2” as following,
“iNOS and COX-2 are pro-inflammatory proteins that are the key enzymes catalyzing the production of NO and PEG2, respectively [21]. To investigate the effect of atraric acid on the production of NO and PGE2, the expressions of iNOS and COX-2 were evaluated by the western blot and flow cytometry. Our results showed that LPS induced significantly increased expressions of iNOS and COX-2 whereas atraric acid resulted in decreased expressions of iNOS and COX-2 in a concentration-dependent manner (Figure 5)”
Q7. Lines 136-138: As shown in Figure 6c the expression of p-IκB was not statistically significant reduced by tartaric acid.
A7: We apologize for the confusion from this data. We missed the ** for statistical significance in figure 6(c) during processing the data. We corrected the figure 6(c) as shown in revised manuscript.
Q8. Lines 159-161: this sentence is not clear “atraric acid treatment…… improved the pathological properties of the organs”.
A8: We correct the whole paragraph in “2.6. Atraric acid exhibited anti-inflammatory effects on LPS-induced endotoxin shock in mice” to clearly explain our results as following,
“Based on in vitro study for anti-inflammatory activity of atraric acid, the in vivo efficacy of atraric acid was evaluated on LPS-induced endotoxin shock in mice. Endotoxin shock dose not only increase the level of cytokines in the blood, but also induces a tissue damage, causing secondary disease [23,24]. The inhibitory effect of atraric acid on the release of inflammatory cytokines from the serum and peritoneal lavage fluid was first investigated. As shown in Figure 7, the LPS-stimulated group significantly induced significantly increased production of pro-inflammatory cytokines in mice whereas atraric acid clearly inhibited the production of cytokines (Figure 7b-e). Next, we performed the hematoxylin and eosin (H&E) staining to analyze the pathological characteristics of organs including kidney, liver and lung. In the LPS group, vasodilation in glomerular atrophy, bleeding, and recruitment of inflammatory cells was observed in kidney, and the similar pathological damages such as vasodilation and bleeding were observed in liver and lung (Figure 7f). In addition, it was observed that the alveolar septa became thicker and the alveoli exuded, leading to the destruction of some alveolar structures in the LPS-stimulated lung. On the contrary, the atraric acid group reduced pathological damages such as vasodilation and bleeding. Taking together, it was confirmed that atraric acid suppresses the inflammatory responses induced by LPS in vitro and in vivo.”
Q9. The discussion is poor and must be improved.; 175-178: this sentence is not clear, “its value….” whose? who is the subject of this sentence?
A9: We corrected the part of discussion as following,
“ There is a great interest in the discovery of biologically active natural products due to their activities with relatively low toxicity as compared to other modalities such as synthetic chemicals [10]. Numerous studies have showed that natural products can be developed as beneficial dietary supplements for health and therapeutic agents against a disease [25]. Lichen is a unique organism that produces biologically active secondary metabolites with various pharmacological effects including antioxidant, anti-cancer and anti-viral activities. In the LPS-stimulated inflammation, toll like recptor-4 (TLR4) and MD-2 in macrophages form heterodimers that recognize a common pattern of LPS molecule [26]. Upstream signaling initiated from the interaction of LPS and TLR4 complex induces the activation of MAPKs/NFkB, which results in overexpression of inflammatory factors [22]. The overexpression of the inflammatory factors including iNOS, COX-2 and pro-inflammatory cytokines including TNF-α, IL-6, and IL-1β induces cellular inflammatory responses and consequently leads in vivo damages such as endotoxin shock (Figure 8) [6,21]. LPS-stimulated macrophage has been generally used as an in vitro model to evaluate anti-inflammatory activity of numerous natural products and synthetic compounds.
The purpose of this study was to discover an anti-inflammatory agent from natural product, which was successfully achieved by isolation and in vitro/in vivo evaluation of secondary metabolites in the natural substance tributary. Atraric acid has been previously reported to have various biological activities such as antibacterial and anti-cancer activity. However, we first isolated the atraric acid from hyterodermia hypoleuca, one of lichens, and report its anti-inflammatory activity in vitro and in vivo. In this work, we first screened several fractions of solvent extracts of hyterodermia hypoleuca by estimating inhibitory activities for the production of inflammatory factors including NO, TNF-α, IL-6, IL-1β, and GM-CSF in LPS-stimulated macrophages, which revealed that the methanol extracts showed the most potent anti-inflammatory activity (Figure 1). Next, we separated and purified atraric acid from the methanol extract using HPLC (Figure 2, S2, Table S1). The isolated atraric acid exhibited a strong inhibition for the production of NO, PGE2, IL-6, IL-1β and GM-CSF in LPS-stimulated RAW264.7 macrophages (Figure 4). Atraric acid also reduced the protein expression of iNOS and COX-2 which are enzymes for the production of inflammatory mediators such as NO and PGE2. In order to prove the anti-inflammatory effect of atraric acid in vivo, atraric acid was evaluated on LPS-induced endotoxin shock in mice. Endotoxin shock was induced by the treatment of LPS (10 mg/kg i.p. injection) in mice, and two concentrations of atraric acid (10 and 30 mg/kg) were inoculated 2 h before and 4 h after LPS inoculation. Excessive cytokines in the blood and peritoneum were observed due to the systematic inflammatory reactions in the LPS-stimulated groups, and also, there were the histological signs of necrosis derived from vasodilation and bleeding of the organs. However, the increased cytokines by LPS were controlled by atraric acid in the atraric acid treatment group, which was supported by the histological staining showing the reduced expansion of capillaries (Figure 7b-f). These in vitro and in vivo data demonstrate that atraric acid can be a promising therapeutic agent against inflammatory diseases.”
Q10. Lines 179-182: this sentence is not clear, perhaps it should be divided into two sentences.
A10: As shown in A9, we corrected the part of discussion.
Q11. Line 188: the data must be show (see above).; Figure 8 is never mentioned in the text.
A11: The explanation for Figure 8 was added in the discussion as show in A9 (Lines: 191-197)

Reviewer 2 Report
In this manuscript (ID# ijms-927727), titled “Atraric acid exhibits anti-inflammatory effect in lipopolysaccharide-stimulated RAW264.7cells and mouse models”, authors, Mun et al, studied the effect atraric acid on lipopolysaccharide-stimulated inflammatory responses in a cell-line and in mice. They concluded that atraric acid has anti-inflammatory effects, which are mediated by inactivation of ERK/NFkB signaling pathway. However, there are several major concerns, which are listed in the following paragraphs:
- The atraric acid used in this study was isolated by authors’ lab using HPLC. The detail information regarding the method of isolation method and the result of structure analysis should be provided.
- Atraric acid used in this study has 90% purity. Please provide the detail information regarding how the purity was determined. What are the other components (about 10%) in the extract? How to exclude the effects induced by non-atraric acid compounds contained in the extract.
- The Cox-2 expression was detected in the cells. To confirm the altered expression of Cox-2 associated with functional changes, the major product of Cox-2, prostaglandin, should be measured.
- The quality of each figure should be improved. Please delete the unnecessary lines in the figures and increase their visibility.
- In the in vivo study using mice, atraric acid was injected subcutaneously, how long the drug can reach to the concentration peak in the plasma; and how long it can distributes into the lung and kidney to generate the protective effects? Therefore, the concentration of atraric acid in the plasma and tissue should be measured. The time-dependent curve should be generated.
- The figure of H&E staining in Fig 7 is not clear enough to see the vascular fluid filtration. Higher amplification microscope lens or immunohistochemistry technique should be used.
- Atraric acid was dissolved in ethyl acetate. Therefore, the ethyl acetate should be used as negative control, instead of PBS.
- What is the target of atraric acid underlying the anti-inflammation effect? The androgen receptor has been identified as the target of atraric acid previously. Is the anti-inflammation effect is due to inhibition of androgen receptors?
Author Response
Q1 .The atraric acid used in this study was isolated by authors’ lab using HPLC. The detail information regarding the method of isolation method and the result of structure analysis should be provided.
A1: A schematic diagram of the extraction method and fractionation was add in supplementary data (Figure S1), and the part of “2.2” in manuscript provides the spectral data including NMR and MS for structural analysis.
Q2. Atraric acid used in this study has 90% purity. Please provide the detail information regarding how the purity was determined. What are the other components (about 10%) in the extract? How to exclude the effects induced by non-atraric acid compounds contained in the extract.
A2: As a result of HPLC analysis, the purity of the atraric acid isolated from the extract is 96.5%. HPLC chromatograms of separated atraric acid was added in supplementary data (Figure S2, Table S1).
Q3. The Cox-2 expression was detected in the cells. To confirm the altered expression of Cox-2 associated with functional changes, the major product of Cox-2, prostaglandin, should be measured.
A3: Depending on your suggestion, PGE2 production was measured by ELISA and added to Figure 4.
Q4. The quality of each figure should be improved. Please delete the unnecessary lines in the figures and increase their visibility.
A4: Following your advice, we changed the figures with improved quality.
Q5. In the in vivo study using mice, atraric acid was injected subcutaneously, how long the drug can reach to the concentration peak in the plasma; and how long it can distributes into the lung and kidney to generate the protective effects? Therefore, the concentration of atraric acid in the plasma and tissue should be measured. The time-dependent curve should be generated.
A5: As mention in your opinion, the subcutaneous injection is an injection method for the medications administered and absorbed slowly, considering the maximum concentration in plasma over time. However, we injected atraric acid by peritoneal injection. Intraperitoneal injection is generally used for the fast distribution compared to intravenous injection, where the drug spreads quickly into the whole body. Therefore, the atraric acid i.p. injection is infused as it will be distributed into the whole blood.
Q6. The figure of H&E staining in Fig 7 is not clear enough to see the vascular fluid filtration. Higher amplification microscope lens or immunohistochemistry technique should be used.
A6: It was confirmed that the existing data are small in size, so that pathological features are not clearly visible. Therefore, the pathological characteristics were adjusted to the size that could be identified and corrected as follows (figure 7). The percentage of photos used is x400.
Q7. Atraric acid was dissolved in ethyl acetate. Therefore, the ethyl acetate should be used as negative control, instead of PBS.
A7: We used Atraric acid suspended in DMSO. EA used for fractionation was completely removed during the next separation and purification.
Q8. What is the target of atraric acid underlying the anti-inflammation effect? The androgen receptor has been identified as the target of atraric acid previously. Is the anti-inflammation effect is due to inhibition of androgen receptors?
A8: Many evidences from different studies show an immuosuppressive role of androgens in different immune cell types [1, 2]. Androgens including testosterone, which is an endogenous ligand of androgen receptor (AR), shows the anti-inflammatory activity on macrophages, which results from androgens-mediated suppression of proinflammatory cytokines such as TNF-α and IL-1β [3, 4]. However, atraric acid has been already reported to be an antagonist of AR [5]. Many studies revealed that AR antagonists, such as bicalutamide and recently reported enzalutamide, antagonize the action of androgens, and thereby leads the cellular senescence and the inhibition of tumor growth in prostate cancer [5, 6]. If atraric acid acts on AR as an antagonist, it can’t lead the anti-inflammatory effects on the Raw264.7 cells, depending on the previously reported studies. Our work clearly showed the anti-inflammatory effect of atraric acid in vitro and in vivo, but it is not clear what the exact mode of action (MOA) for anti-inflammatory activity of atraric acid is. We are currently doing the further study for atraric acid, and will report the exact MOA of atraric acid for its anti-inflammatory effect in Int. J. Mol. Sci. in future.
References in A8
[1] Melanie R, G.; Trine N. J. Androgen-Induced Immunosuppression. Front Immunol. 2018, 9, 794.
[2] Isabel, B.; Maria, D.; Gunhild, A.; Melanie, J.; Sonja, L.; Influence of Androgens on Immunity to Self and Foreign: Effects on Immunity and Cancer. Front. Immunol., 2020, 11, 1184.
[3] Mireya, B.; Mason, S.; Nicola, H.; Androgen and Androgen Receptors as Regulators of Monocyte and Macrophage Biology in the Healthy and Diseased Lung. Front. Immunol. 2020, 11, 1698.
[4] Abhishek, T.; Joana, D.; Trine, J.; Suppressive effects of androgens on the immune system. Cellular Immunology. 2015, 294, 87-94
[5] Hessenkemper, W.; Roediger, J.; Bartsch, S.; Houtsmuller, A.B.; Van-royen, M.E.; Petersen, I.; Grimm, M.O.; Baniahmad, A. A natural androgen receptor antagonist induces cellular senescence in prostate cancer cells. Mol Endocrinol. 2014, 28(11), 1831‒1840.
[6] Christine, H.; Thomas, B.; Arnout, V.; Stefan, P.; Hedrik, P.; Steven, J.; Frank, C.; Androgen receptor antagonists for prostate cancer therapy. Endocrine-Related Cancer. 2014, 21, 105–118.

Round 2
Reviewer 1 Report
The English must be improved. Although understandable, the language needs to be improved in numerous sentences such as that reported on lines 167-169.
Author Response
Response of Reviewer
#Open Reviewer 1
Comments and Suggestions for Authors
The English must be improved. Although understandable, the language needs to be improved in numerous sentences such as that reported on lines 167-169.
A : We have been corrected by professional proofreaders (Editage, https://www.editage.co.kr) for improved English.
We corrected the part as following, “As shown in Figure 7, the LPS-stimulated group of mice exhibited significantly induced significantly increased production of pro-inflammatory cytokines, in mice whereas for ataric acid-treated mice, clearly inhibited the production of pro-inflammatory cytokines was inhibited (Figure 7b-e).”(Lines: 174-176)
Reviewer 2 Report
The manuscript has been improved and no further concern.
Author Response
#Open Reviewer 2
Comments and Suggestions for Authors
The manuscript has been improved and no further concern.
A : We would like to thank the reviewers for their insightful comments (minor revision) to our manuscript, entitled “Atraric acid exhibits anti-inflammatory effect in lipopolysaccharide-stimulated RAW264.7cells and mouse models”. To best respond to the reviewer’s comments, we entrusted the correction of English from Editage (https://www.editage.co.kr).
Round 3
Reviewer 1 Report
The manuscript has been excellently revised and improved. I have no further criticisms